# Association of the incidence of atopic dermatitis until 3 years old with climate conditions in the first 6 months of life: Japan Environment and Children's Study (JECS)

**Hiroshi Yokomichi**[1]*, **Mie Mochizuki**[2], **Ryoji Shinohara**[3], **Megumi Kushima**[3], **Sayaka Horiuchi**[3], **Reiji Kojima**[1], **Tadao Ooka**[1], **Yuka Akiyama**[1], **Kunio Miyake**[1], **Sanae Otawa**[3], **Zentaro Yamagata**[1,3], **on behalf of the Japan Environment and Children's Study Group**[¶]

1 Department of Health Sciences, University of Yamanashi, Chuo City, Yamanashi, Japan, 2 Department of Pediatrics, University of Yamanashi, Chuo City, Yamanashi, Japan, 3 Center for Birth Cohort Studies, University of Yamanashi, Chuo City, Yamanashi, Japan

¶ The members of the Japan Environment and Children's Study Group are listed in the Acknowledgments.
* hyokomichi@yamanashi.ac.jp

**Data Availability Statement:** Data are available on reasonable request. Data are unsuitable for public deposition because of ethical restrictions and the

## Abstract

### Objective

To determine the climate conditions that affect the incidence of atopic dermatitis from infancy to 3 years old.

### Study design

We analyzed 100,303 children born from 2011 to 2014 for follow-up until 3 years old using cohort data from the Japan Environment and Children's Study. The study included 15 Regional Centers, including 19 prefectures across Japan. We used meteorological data of the Japan Meteorological Agency. We calculated the hazard ratio (HR) of the standard deviation and low vs. high mean values of several climate conditions in children in their first 6 months of life to determine the incidence of atopic dermatitis.

### Results

The Kaplan–Meier curve showed that children born in the months of October to December had the highest incidence of atopic dermatitis. Among climate conditions, a one standard deviation increase in the temperature (HR = 0.87), minimum temperature (HR = 0.87), and vapor pressure (HR = 0.87) showed the lowest HRs for the incidence of atopic dermatitis. These results were confirmed by an analysis by strata of the birth season. A low vapor pressure (HR = 1.26, p<0.0001) and the combination of a low mean temperature or low mean minimum temperature and low vapor pressure (HR = 1.26, p<0.0001) were associated with the highest incidence of atopic dermatitis. These results were consistent when they were adjusted for a maternal and paternal history of allergy and the prefecture of birth.

legal framework of Japan. The Act on the Protection of Personal Information (Act No. 57 of 30 May 2003, amendment on 9 September 2015) prohibits public deposition of the data containing personal information. Ethical Guidelines for Medical and Health Research Involving Human Subjects enforced by the Japan Ministry of Education, Culture, Sports, Science and Technology and the Ministry of Health, Labor and Welfare also restrict the open sharing of epidemiologic data. All inquiries about access to data should be sent to: jecsen@nies.go.jp. The person responsible for handling enquiries sent to this email address is Shoji F. Nakayama, JECS Program Office, National Institute for Environmental Studies.

**Funding:** This study was funded by the Ministry of the Environment, Japan (to ZY, grant no: none. https://www.env.go.jp/chemi/ceh/). The funder had no role in study design, data collection and analysis, decision to publish, or preparation of the manuscript. The findings and conclusions of this article are solely the responsibility of the authors and do not represent the official views of the above government.

**Competing interests:** The authors have declared that no competing interests exist.

## Conclusion

Among climate conditions, a low vapor pressure is the most strongly associated with a high incidence of atopic dermatitis. Measuring vapor pressure may be useful for preventing atopic dermatitis.

## Introduction

Globally, approximately 10%–20% of children are affected by atopic dermatitis (AD) [1, 2]. Investigation of the cause of AD is necessary to help relieve patients' symptoms and reduce the number of patients. To date, many genetic and environmental factors are considered to be candidates for causing AD [3, 4]. Among the candidates of environmental risk factors [5, 6], seasonal climate conditions, chemical irritants, bacterial colonization, psychological stress [5], and birth month [7] are the main factors.

Our previous study showed that birth from April to June had the smallest risk of AD, and birth from October to December had the greatest risk of AD [7]. Theoretically, a genetic predisposition for AD does not affect the association between exposure of the birth month and the outcome of AD as a confounding factor. Therefore, this association appears to be affected by an environmental factor(s). The particular environmental factor involved in AD may be the climate condition because the difference in the risk of AD between April and June and that between October and December appears to be due to the season when children are exposed. In our previous study, Kaplan–Meier curves suggested that the amount of AD risk by birth season does not change much from 6 months to 3 years of age [7]. This finding suggests that the determinants of developing AD up to 3 years of age could be environmental factors from birth to 6 months of age.

Relating birth cohort data [8–11] and meteorological data [12] may help answer the question of how the birth month is associated with the incidence of AD. The reason for the association between the birth month and the prevalence of AD needs to be determined. Several meteorological measures represent features of seasonality in which newborns are exposed. Therefore, this study aimed to examine which climate element for birth in spring and autumn is associated with the incidence of AD.

## Materials and methods

### Ethics statement

The Japan Environment and Children's Study (JECS) protocol was reviewed and approved by the Ministry of the Environment's Institutional Review Board on Epidemiological Studies and by the Ethics Committees of all participating institutions. The study was performed in accordance with the ethical guidelines and regulations of the Declaration of Helsinki. All participants and parents or guardians of the children provided written informed consent before participating in the study.

### Measures

Details on the JECS cohort are published elsewhere [8]. Approximately 100,000 expecting mothers who lived in the designated study areas were recruited over 3 years from January 2011. Exposure to environmental factors was assessed by chemical analyses of biospecimens and household environment measurements using monitoring data and questionnaires. One of

the JECS' priority outcomes was allergic diseases [8]. We recruited expecting mothers from January 2011 to March 2014 in 15 Regional Centers covering 19 prefectures across Japan [8]. We used the JECS data "jecs-ta-20190930-qsn" for answers to questionnaires, which were sent by post or provided to caregivers when their children were aged 6 months, 1 year, 1.5 years, 2 years, and 3 years. We asked if physicians had diagnosed the children with AD.

We collected climate condition data by prefecture and month from the Japan Meteorological Agency website [12]. We downloaded data of the monthly mean temperature, maximum temperature, minimum temperature, temperature difference, precipitation amount, sunshine duration, sunshine percentage, solar radiation quantity, vapor pressure, humidity, wind velocity, and cloud cover from February 2011 to May 2015. We regarded the 6-month mean climate condition value as the environment to which newborns were exposed. We chose this time because, in our previous study, the environment of the first 6 months after birth had the potential to determine the environmental risk of AD [7]. An example of this time period is neonates who were born in February 2011 were regarded to be exposed to climate conditions from February to July 2011.

From February 2011 to May 2015, we calculated the overall means of data of the monthly means of climate condition values from February 2011 to May 2015. These values were used as cut-off values for categorizing children as being exposed to a high/low climate condition. Using this categorization, the climate condition in the child's residential area over 6 months beginning with the child's birth month was compared with the cut-off values.

We used a maternal and paternal history of asthma, allergic rhinitis, pollen allergy, AD, allergic conjunctivitis, and/or food allergy for considering a child's genetic predisposition to AD as covariates in a random effects model. The father and mother were asked individually about a history of allergic diseases to determine their experience of each diagnosed disease by a physician.

## Statistical analyses

The 12 months of the year were categorized into four seasons (i.e., starting with January to March). We constructed a Kaplan–Meier curve of incident AD by the birth season. Using the Cox proportional model, we calculated the hazard ratio (HR) for a one standard deviation (SD) increase in the climate condition. We conducted this analysis with 6-month and 3-month means of the climate condition. Because 6-month means were more strongly associated with the incidence of AD (Table 1), we also calculated HRs for 6-month means of the climate condition by strata of the birth season. Additionally, we calculated HRs of low vs. high values of the climate condition for 6 months from birth. In this analysis, because we found a large difference in the incidence of AD among temperature, minimum temperature, and vapor pressure, we calculated HRs of combinations of high/low mean/maximum/minimum temperature and high/low vapor pressure.

We then calculated this HR with adjustment for a maternal and paternal history of allergy and birthplace (prefectures) in the random effect model. In survival analyses, data from participants who were lost to follow-up (n = 20,652) or those who did not develop AD until 3 years of age (n = 69,001) were considered as being censored. Finally, we plotted the latitude and the accumulated incidence of AD at 3 years of age among the birthplaces. Although there were 19 different prefectures, a small subset of data in one prefecture was treated as belonging to data of another prefecture. Therefore, we plotted the latitude for 18 prefectural meteorological observatories. In these plots, we calculated Pearson's correlation coefficients and p values. In this analysis of our ecological study, if there was a climate condition-associated incidence of AD, we attempted to determine whether it was simply attributable to latitude. We conducted

**Table 1. HRs (95% confidence intervals) of the incidence of atopic dermatitis for a one standard deviation increase in the mean climate condition from birth to 3 or 6 months.**

| Climate condition | Mean (SD) | HR for 1 SD, 6 months | HR for 1 SD, 3 months |
|---|---|---|---|
| Temperature | 14. 9 (5.7) | 0.87* (0.86, 0.89) | 0.91* (0.89, 0.92) |
| Maximum temperature | 19.3 (5.9) | 0.88* (0.86, 0.89) | 0.91* (0.89, 0.92) |
| Minimum temperature | 11.1 (5.9) | 0.87* (0.85, 0.89) | 0.91* (0.89, 0.92) |
| Temperature difference | 8.2 (1.3) | 1.04* (1.02, 1.06) | 1.01 (0.99, 1.03) |
| Precipitation amount | 137 (64) | 0.91* (0.89, 0.93) | 0.94* (0.92, 0.96) |
| Sunshine duration | 167 (26) | 0.98* (0.96, 0.99) | 0.95* (0.93, 0.97) |
| Sunshine percentage | 45.7 (6.8) | 1.03* (1.01, 1.05) | 1.01 (0.99, 1.03) |
| Solar radiation quantity | 13.3 (2.8) | 0.94* (0.92, 0.96) | 0.91* (0.89, 0.93) |
| Vapor pressure | 13.5 (4.9) | 0.87* (0.85, 0.88) | 0.91* (0.89, 0.93) |
| Atmospheric pressure | 1,007 (9.9) | 0.98* (0.96, 0.999) | 0.99 (0.97, 1.01) |
| Humidity | 68.5 (5.4) | 0.92* (0.90, 0.94) | 0.96* (0.94, 0.98) |
| Wind velocity | 3.02 (0.93) | 0.95* (0.93, 0.97) | 0.94* (0.92, 0.96) |

*$p < 0.05$. SD, standard deviation; HR, hazard ratio.

all statistical analyses using SAS statistical software version 9.4 (SAS Institute, Cary, NC, USA). A p value of <0.05 in two-tailed tests was considered to indicate a significant difference.

## Results

The data of 100,303 children were analyzed. By the ages of 6 months, 1 year, 1.5 years, 2 years, and 3 years, accumulated numbers of 1,715 of 100,303 children, 4,505 of 90,549 children, 7,030 of 86,934 children, 8,558 of 83,859 children, and 10,650 of 79,651 children, respectively, had developed AD. Fig 1 shows the association between the birth season and the incidence of AD to 3 years of age. From 6 months to 3 years of age, birth between April and June had the lowest incidence of AD, and birth between October and December had the highest incidence of AD.

Table 1 shows the HRs of AD for a one SD increase in the climate conditions. Among the climate conditions, HRs were significantly lower for a one SD increase in the 6-month mean temperature (HR = 0.87, p<0.0001), maximum temperature (HR = 0.88, p<0.0001), minimum temperature (HR = 0.87, p<0.0001), and vapor pressure (HR = 0.87, p<0.0001). Table 2 shows the HRs of a one SD increase in the climate condition by the birth season. In this analysis, a higher temperature, higher maximum temperature, higher minimum temperature, and higher vapor pressure also showed significantly lower HRs of AD (all p<0.0001). Among the seasons, birth between April and June showed the lowest HRs for a higher temperature, higher maximum temperature, higher minimum temperature, and higher vapor pressure (HRs = 0.64–0.68).

Table 3 shows the HRs of AD for a low vs. high climate condition value. A low temperature (HR = 1.23, p<0.0001), low maximum temperature (HR = 1.21, p<0.0001), low minimum temperature (HR = 1.23, p<0.0001), and low vapor pressure (HR = 1.26, p<0.0001) showed significantly higher HRs than those for a high temperature, high maximum temperature, high minimum temperature, and high vapor pressure, respectively. A low vapor pressure only, the combination of a low temperature and a low vapor pressure, and the combination of a low minimum temperature and a low vapor pressure had the highest HRs (1.26) of AD.

S1 Table shows the HRs of AD for high climate condition values with adjustment for a maternal and paternal history of allergy and birthplace. A low temperature, low maximum temperature, low minimum temperature, and low vapor pressure had significantly higher

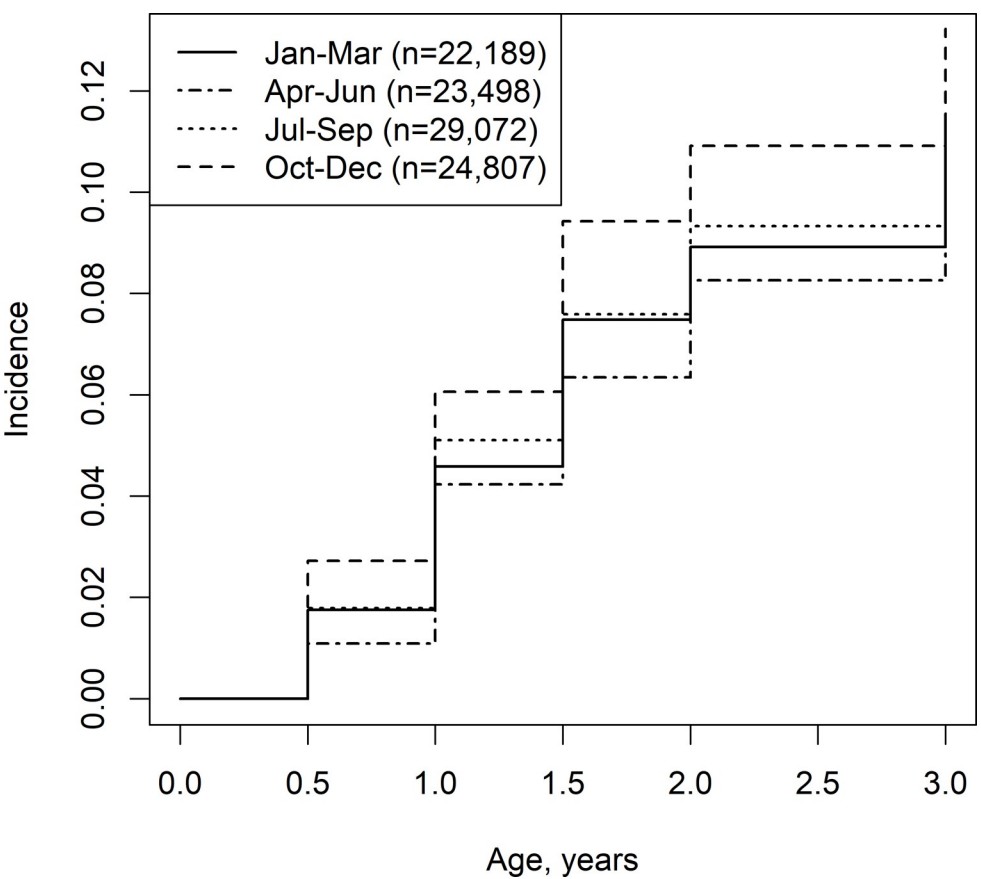

**Fig 1. Birth season and the incidence of atopic dermatitis up to 3 years of age.**

**Table 2. Crude HRs of the incidence of atopic dermatitis for a one standard deviation increase in the mean climate condition from birth to 6 months by birth season.**

| Climate condition | HR for 1 SD | HR for 1 SD | HR for 1 SD | HR for 1 SD |
|---|---|---|---|---|
| | Birth from January to March | Birth from April to June | Birth from July to September | Birth from October to December |
| Temperature | 0.83* | 0.64* | 0.86* | 0.86* |
| Maximum temperature | 0.85* | 0.68* | 0.87* | 0.87* |
| Minimum temperature | 0.82* | 0.65* | 0.85* | 0.85* |
| Temperature difference | 1.06* | 1.04* | 1.06* | 1.01 |
| Precipitation amount | 0.96 | 0.95* | 0.92* | 0.89* |
| Sunshine duration | 0.96 | 0.96 | 1.01 | 1.01 |
| Sunshine percentage | 0.995 | 0.95 | 1.03 | 1.02 |
| Solar radiation quantity | 0.95 | 0.91* | 0.94 | 1.01 |
| Vapor pressure | 0.83* | 0.65* | 0.83* | 0.79* |
| Atmospheric pressure | 0.94* | 0.98 | 0.96* | 0.98 |
| Humidity | 0.95* | 0.95* | 0.90* | 0.98 |
| Wind velocity | 0.91* | 0.97 | 0.94* | 0.93* |

*p<0.05. SD, standard deviation; HR, hazard ratio.

**Table 3. HRs (95% confidence intervals) of the incidence of atopic dermatitis for a low vs. high mean climate condition from birth to 6 months.**

| Low vs. high value | Cut-off value for high or low | HR |
|---|---|---|
| Temperature, ˚C | 15.8 | 1.23 (1.18, 1.28) |
| Maximum temperature, ˚C | 20.2 | 1.21 (1.16, 1.26) |
| Minimum temperature, ˚C | 12.1 | 1.23 (1.19, 1.29) |
| Precipitation amount, mm | 148 | 1.20 (1.15, 1.25) |
| Sunshine duration, hours | 166 | 1.05 (1.01, 1.09) |
| Sunshine percentage, % | 45 | 0.95 (0.92, 0.99) |
| Solar radiation quantity, MJ/m$^2$ | 13.8 | 1.13 (1.08, 1.17) |
| Vapor pressure, hPa | 14.4 | 1.26 (1.21, 1.31) |
| Atmospheric pressure, hPa | 1,006 | 0.85 (0.81, 0.90) |
| Humidity, % | 69 | 1.19 (1.15, 1.24) |
| Wind velocity, m/s | 3.1 | 0.90 (0.86, 0.93) |
| Low temperature and low vapor pressure | | 1.26 (1.21, 1.31) |
| Low temperature and high vapor pressure | | 0.74 (0.59, 0.93) |
| High temperature and low vapor pressure | | 1.14 (1.03, 1.26) |
| High temperature and high vapor pressure | | Reference |
| Low maximum temperature and low vapor pressure | | 1.25 (1.20, 1.30) |
| Low maximum temperature and high vapor pressure | | 0.78 (0.65, 0.93) |
| High maximum temperature and low vapor pressure | | 1.20 (1.09, 1.32) |
| High maximum temperature and high vapor pressure | | Reference |
| Low minimum temperature and low vapor pressure | | 1.26 (1.21, 1.31) |
| Low minimum temperature and high vapor pressure | | 0.83 (0.64, 1.09) |
| High minimum temperature and low vapor pressure | | 1.17 (1.06, 1.30) |
| High minimum temperature and high vapor pressure | | Reference |

HR, hazard ratio.

adjusted HRs, which ranged from 1.17 to 1.19, than their corresponding high values (all p<0.001). A low temperature only, the combination of a low temperature and a low vapor pressure, the combination of a low maximum temperature and a low vapor pressure, and the combination of a low minimum temperature and a low vapor pressure showed the highest adjusted HRs (1.19) (all p<0.001).

S1 Fig shows a scatter plot of 18 prefectures by latitude and the accumulated incidence of AD until 3 years of age for an ecological study. Pearson's correlation coefficient was 0.409 and the p value was 0.092. We did not find a significant relationship between the latitude and the incidence of AD in this ecological study.

## Discussion

When we investigated the association of AD with the climate condition, while avoiding confounding, a low vapor pressure was most strongly associated with a higher incidence of AD in childhood. Temperature, maximum temperature, and minimum temperature were also associated with a higher incidence of AD.

This study suggested that the risk of AD by birth month was determined by 6 months of age (Fig 1). Therefore, we considered that season-associated environmental factors should affect the risk of AD. We compared the risk of AD among SDs of climate conditions (Table 1). This comparison suggested that any temperature and vapor pressure could be candidates of environmental risk factors for AD. Additionally, the variability in the mean climate condition

from birth to 6 months was more strongly associated with the incidence of AD than that from birth to 3 months. In Japan, temperature and vapor pressure rise from spring to summer, and they fall from autumn to winter [7]. Therefore, the birth month could confound the association of the risk of AD with temperature and vapor pressure. To avoid confounding, we calculated HRs of climate conditions by strata of the birth season (Table 2). Our results suggested that temperature and vapor pressure affected the risk of AD separately from the birth month. Additionally, a decreased risk of AD with birth from April to June may be greatly affected by exposure to a higher temperature and a higher vapor pressure.

To understand the environmental risk factors, we calculated HRs of low vs. high climate conditions (Table 3). We found higher HRs (1.23–1.26) for temperature or vapor pressure in the current study than that (HR = 1.20) in our previous study when we compared birth from October to December with birth from April to June [7]. We could not initially determine which was the most important risk factor among temperature, maximum temperature, minimum temperature, and vapor pressure. Therefore, we calculated all HRs of these conditions for low vs. high values. We also investigated the influence of the combination of a low/high temperature and a low/high vapor pressure on the risk of AD. Although a low temperature and a low vapor pressure were associated with an increased incidence of AD, the combination of a low temperature and a low vapor pressure did not have a synergistic influence on this incidence (HR = 1.26). However, a low vapor pressure only showed the same HR (1.26). Therefore, vapor pressure was considered as an important climate condition for developing AD.

Few studies have investigated the association between the climate condition during the neonatal period and the risk of AD. A previous study investigated the association between temperature and ambulatory visits with AD in the United States, and showed that increased temperature increased the likelihood of office visits for AD [13]. However, in the investigation of calendar months, there was no clear effect of seasonality on office visits for AD. Ecological data suggested an inverse association between the mean annual temperature and the prevalence of AD in children, and showed no clear association between humidity and the prevalence of AD [14]. Our study, which used relatively individual data, also showed an association between a higher temperature and a lower incidence of AD. A high temperature enables air to contain more water, and therefore this could increase vapor pressure. The present analysis also showed that a higher vapor pressure decreased the incidence of AD.

There are inconsistent ecological data of the association between the latitude and the incidence of AD [14, 15]. An Australian report suggested that a higher incidence of eczema was associated with a higher latitude of residence (odds ratio = 1.90 for south vs. north regions) [16]. In our study, there was a slight correlation between the latitude and the accumulated incidence of AD, although this was not significant (S1 Fig). Low latitude areas receive solar irradiation, which produces vitamin D. Because vitamin D supplementation may help ameliorate the severity of AD [17], residing at a low latitude may prevent AD. Vapor pressure and temperature are likely to be high if the latitude of the residence is low. These previous studies [14–16] and our ecological analyses were not able to determine the climate condition responsible for the risk of AD.

AD is caused by the interplay of several factors [18]. Skin dryness in winter [19] may partly explain the high incidence of AD in children born between October and December. Vapor pressure is due to the pressure of water vapor, which is determined by the temperature and volume of gasiform water in a unit volume of air. In contrast, humidity (or relative humidity), which is frequently announced in the weather forecast, is the percentage of water volume of the maximum water volume that a unit volume of air can contain at a particular temperature. Therefore, vapor pressure and humidity are strongly correlated. These definitions mean that vapor pressure rather than humidity can better indicate the moisture in air and may be directly

associated with skin moisture. This may be the reason why vapor pressure was more strongly associated with the incidence of AD than humidity in this study. This is in line with our finding that newborns delivered between October and December had the highest incidence of AD (Fig 1) because they were surrounded by a low vapor pressure in the first 6 months (winter) from birth.

If vapor pressure rather than humidity is important for developing AD, we could use this information to prevent child AD. If neonates are born in autumn to winter, they experience a low vapor pressure for the first 6 months of life. Parents could prepare a humidifier for their neonate to decrease this disease risk. They could also pay attention to the vapor pressure value. To prevent AD, a home electrical appliance that shows vapor pressure rather than humidity could be more useful for patients with AD. Because humidity is calculated from the saturated vapor pressure and real vapor pressure, this type of appliance should be able to be manufactured.

Parents need to treat skin dryness in winter when vapor pressure falls for infants who are born between October and December. In a season or area of dry air, effort should be made to achieve retention of moisture. A meta-analysis showed that emollient use in infants younger than 6 months was significantly effective for preventing AD, while there were conflicting results on its effectiveness in infants aged between 6 and 12 months [20]. Because a low vapor pressure was a risk factor for AD in our study, protecting the skin from dry air should decrease the risk of AD.

Previous studies investigated latitude [16], humidity [21], or sunshine [22] as a risk or preventive factor for AD. Each climate condition changes along with other climate conditions. Therefore, the climate conditions confound each other. The present study showed that vapor pressure may play a principal role in the risk of AD. To the best of our knowledge, no studies have shown that vapor pressure is a risk factor for AD. We propose that measuring vapor pressure could be useful for the prevention of AD.

Our study had the following limitations. First, the indoor condition was not measured in this study. We could not exclude the effect of a household environment with the use of cleaners, bleaches, and detergents. Second, our results are limited because we did not detect *Staphylococcus aureus* in the skin [23], which could contribute to an increased risk of AD. Third, dichotomizations of a long/short sunshine duration and high/low humidity were determined by single cut-off values. Observational results could have varied, depending on the cut-off values. Fourth, the incidence of AD was reported by caregivers based on a physician's diagnosis. Therefore, there could have been recall bias of the caregivers. Fifth, physicians who diagnosed children included specialists of AD and non-specialists. Non-specialist physicians might have underdiagnosed AD in infancy because this diagnosis can cause stigma to children and caregivers [24]. Sixth, short-term factors, such as yellow sand from China and seasonal pollen in the air, were not considered in this study. Studies have suggested that these factors may worsen dermatitis [25, 26].

## Conclusions

Among climate conditions, such as temperature and relative humidity, a low vapor pressure is the most strongly associated with a high incidence of AD in Japan. Paying attention to vapor pressure may reduce the incidence of AD in neonates who are born between October and December.

## Supporting information

**S1 Table. Adjusted HRs (95% confidence intervals) of atopic dermatitis for a low vs. high mean climate condition from birth to 6 months.**
(DOCX)

**S1 Fig. Scatter plot of the latitude and the accumulated incidence of atopic dermatitis at 3 years of age.** Owing to the small number of participants, data from Shiga Prefecture were included with those of the neighboring Kyoto Prefecture.
(TIF)

## Acknowledgments

We express our gratitude to all study participants and co-operating healthcare providers who supported the JECS. We thank Ellen Knapp, PhD, from Edanz for editing a draft of this manuscript.

The members of the JECS Group as of 2021 are as follows: Michihiro Kamijima (principal investigator, Nagoya City University, Nagoya, Japan, jecsen@nies.go.jp), Shin Yamazaki (National Institute for Environmental Studies, Tsukuba, Japan), Yukihiro Ohya (National Center for Child Health and Development, Tokyo, Japan), Reiko Kishi (Hokkaido University, Sapporo, Japan), Nobuo Yaegashi (Tohoku University, Sendai, Japan), Koichi Hashimoto (Fukushima Medical University, Fukushima, Japan), Chisato Mori (Chiba University, Chiba, Japan), Shuichi Ito (Yokohama City University, Yokohama, Japan), Hidekuni Inadera (University of Toyama, Toyama, Japan), Takeo Nakayama (Kyoto University, Kyoto, Japan), Hiroyasu Iso (Osaka University, Suita, Japan), Masayuki Shima (Hyogo College of Medicine, Nishinomiya, Japan), Hiroshige Nakamura (Tottori University, Yonago, Japan), Narufumi Suganuma (Kochi University, Nankoku, Japan), Koichi Kusuhara (University of Occupational and Environmental Health, Kitakyushu, Japan), and Takahiko Katoh (Kumamoto University, Kumamoto, Japan).

## Author Contributions

**Conceptualization:** Hiroshi Yokomichi.

**Formal analysis:** Hiroshi Yokomichi.

**Funding acquisition:** Zentaro Yamagata.

**Investigation:** Hiroshi Yokomichi, Mie Mochizuki.

**Methodology:** Hiroshi Yokomichi.

**Project administration:** Ryoji Shinohara, Megumi Kushima, Sayaka Horiuchi, Sanae Otawa, Zentaro Yamagata.

**Supervision:** Ryoji Shinohara, Zentaro Yamagata.

**Writing – original draft:** Hiroshi Yokomichi, Mie Mochizuki.

**Writing – review & editing:** Hiroshi Yokomichi, Ryoji Shinohara, Megumi Kushima, Sayaka Horiuchi, Reiji Kojima, Tadao Ooka, Yuka Akiyama, Kunio Miyake, Sanae Otawa, Zentaro Yamagata.

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
