## [Decision Letter · Decision Letter 0]

31 Mar 2022

PONE-D-22-05326Association of the incidence of atopic dermatitis until 3 years old with climate conditions in the first 6 months of life: Japan Environment and Children’s Study (JECS)PLOS ONE

Dear Dr. Yokomichi,

Thank you for submitting your manuscript to PLOS ONE. After careful consideration, we feel that it has merit but does not fully meet PLOS ONE’s publication criteria as it currently stands. Therefore, we invite you to submit a revised version of the manuscript that addresses the points raised during the review process.

We look forward to receiving your revised manuscript.

Kind regards,

Svetlana P. Chapoval

Academic Editor

PLOS ONE

Journal Requirements:

3. "One of the noted authors is a group or consortium [the Japan Environment and Children’s Study Group]. In addition to naming the author group, please list the individual authors and affiliations within this group in the acknowledgments section of your manuscript. Please also indicate clearly a lead author for this group along with a contact email address.

Additional Editor Comments:

1. Please clarify if a vapor pressure is equal to atmospheric humidity.

2. How the effect of a household environment (use of cleaners, bleaches, detergents, etc.) was excluded from the study?

3. Was a barometric pressure included in the meteorologic parameters assessed? What about an effect of a short-term pollution?

4. Materials and Methods. Why Kaplan-Meier curves were used in statistical analysis of the obtained data? Specify what chemical analyses of bio-speciments included, what bio-speciments were used, and how such analyses were performed. Same for household environmental measurements. What kind of computational stimulation was used and how? Why a Cox proportional mode was used and what it allowed to achieve? How a latitude was plotted? Why Pearson’s correlation coefficients were evaluated, what information they provide for your study? Two-sided p values mean a two-tailed test was used?

5. Results. Table 1 shows no statistics because there was no difference between the parameters? What does Ref mean as a reference in Table 3.

6. Discussion. Be specific why any temperature (Low, high?) and vapor pressure are associated with a higher incidence of AD. Is a high heat or cold temperature associated with a likelihood of office visits for AD?

7. Conclusion. Make also a statement on the results on temperature and humidity association with higher incidences of AD.

Reviewers' comments:

Reviewer's Responses to Questions

**Comments to the Author**

1. Is the manuscript technically sound, and do the data support the conclusions?

Reviewer #1: Yes

2. Has the statistical analysis been performed appropriately and rigorously? 

Reviewer #1: I Don't Know

3. Have the authors made all data underlying the findings in their manuscript fully available?

Reviewer #1: Yes

4. Is the manuscript presented in an intelligible fashion and written in standard English?

Reviewer #1: Yes

5. Review Comments to the Author

Reviewer #1: This manuscript follows up on prior results indicating that in Japanese individuals a birth date between October to December correlates with increased risk for atopic dermatitis. The current work tries to tease out the features of seasonality that are responsible. The work is interesting and deserves publication.

Specific comments:

-The authors should state limitations of their conclusions in more detail.

-The sentence about a randomized controlled trial showing prevention of AD by use of moisturizer in infants <1year needs to be modified, reflecting the fact that there are multiple studies on this subject with conflicting results.

6. PLOS authors have the option to publish the peer review history of their article (what does this mean?). If published, this will include your full peer review and any attached files.

Reviewer #1: No

---

## [Author Response · Author response to Decision Letter 0]

13 Apr 2022

Responses to the Editor and reviewer

As requested, we have prepared a revised version of our manuscript based on the comments and suggestions received. We hope that these revisions have sufficiently addressed the Editor’s and reviewer’s concerns. Our point-by-point responses are included below each comment (in italics), with line numbers indicating the relevant changes in the revised manuscript. We extend our sincere thanks to the Editor and reviewer for all the helpful comments provided. 

Journal requirements

We thank the Editor for the suggestion. We have edited our manuscript to meet PLOS ONE’s style requirements. 

There is no grant number available for the funding. This information will be mentioned at submission. 

3. One of the noted authors is a group or consortium [the Japan Environment and Children’s Study Group]. In addition to naming the author group, please list the individual authors and affiliations within this group in the acknowledgments section of your manuscript. Please also indicate clearly a lead author for this group along with a contact email address. 

We thank the Editor for the suggestion. We have added the names of the JECS group to the Acknowledgments section. We have also indicated the contact e-mail address of the principal investigator. 

Line 273: “The members of the JECS Group as of 2022 are as follows: Michihiro Kamijima (principal investigator, Nagoya City University, Nagoya, Japan, jecsen@nies.go.jp), Shin Yamazaki (National Institute for Environmental Studies, Tsukuba, Japan), Yukihiro Ohya (National Center for Child Health and Development, Tokyo, Japan), Reiko Kishi (Hokkaido University, Sapporo, Japan), Nobuo Yaegashi (Tohoku University, Sendai, Japan), Koichi Hashimoto (Fukushima Medical University, Fukushima, Japan), Chisato Mori (Chiba University, Chiba, Japan), Shuichi Ito (Yokohama City University, Yokohama, Japan), Hidekuni Inadera (University of Toyama, Toyama, Japan), Takeo Nakayama (Kyoto University, Kyoto, Japan), Hiroyasu Iso (Osaka University, Suita, Japan), Masayuki Shima (Hyogo College of Medicine, Nishinomiya, Japan), Hiroshige Nakamura (Tottori University, Yonago, Japan), Narufumi Suganuma (Kochi University, Nankoku, Japan), Koichi Kusuhara (University of Occupational and Environmental Health, Kitakyushu, Japan), and Takahiko Katoh (Kumamoto University, Kumamoto, Japan).” 

We have reviewed the references to ensure that they are complete and correct. 

Responses to additional comments from the Editor

1. Please clarify if a vapor pressure is equal to atmospheric humidity.

We thank the Editor for the comment. Atmospheric humidity can be specified by the partial pressure of water vapor (e, in hPa), vapor density (g m−3), specific humidity (q, g/g of moist air), or relative humidity (RH = 100e/es) (https://www.sciencedirect.com/topics/earth-and-planetary-sciences/atmospheric-humidity). To avoid confusion of the readers, we used the term “vapor pressure”, and have emphasized the difference between vapor pressure and relative humidity in the Discussion section. 

Line 226: “Vapor pressure is due to the pressure of water vapor, which is determined by the temperature and volume of gasiform water in a unit volume of air. In contrast, humidity (or relative humidity), which is frequently announced in the weather forecast, is the percentage of water volume of the maximum water volume that a unit volume of air can contain at a particular temperature.”

2. How the effect of a household environment (use of cleaners, bleaches, detergents, etc.) was excluded from the study? 

We appreciate the Editor’s insightful comment. Because the household environment was not sufficiently examined, and we could not exclude the effect of this environment, this was a limitation to this study. We have added the following text to the manuscript:

Line 254: “Our study had the following limitations. First, the indoor condition was not measured in this study. We could not exclude the effect of the household environment with the use of cleaners, bleaches, and detergents.”

3. Was a barometric pressure included in the meteorologic parameters assessed? What about an effect of a short-term pollution? 

We included mean barometric pressure as mean atmospheric pressure in the tables. The association between a one standard deviation increase in mean atmospheric pressure and the incidence of atopic dermatitis (AD) was small (0.98, 95% CI: 0.96, 0.999 in Table 1; 0.94, 0.98 in Table 2). Therefore, we did not consider atmospheric pressure as a risk factor for AD. 

Short-term factors, such as yellow sand from China and seasonal pollen in the air, were not considered in this study. This information has been added as another limitation. 

Line 263: “Sixth, short-term factors, such as yellow sand from China and seasonal pollen in the air, were not considered in this study. Studies have suggested that these factors may worsen dermatitis [26, 27].” 

4. Materials and Methods. Why Kaplan-Meier curves were used in statistical analysis of the obtained data? Specify what chemical analyses of bio-speciments included, what bio-speciments were used, and how such analyses were performed. Same for household environmental measurements. What kind of computational stimulation was used and how? Why a Cox proportional model was used and what it allowed to achieve? How a latitude was plotted? Why Pearson’s correlation coefficients were evaluated, what information they provide for your study? Two-sided p values mean a two-tailed test was used? 

We thank the editor for the questions. The Kaplan–Meier curve in Fig 1 was used to illustrate the difference in the incidence of AD from the age of 6 months to 3 years among birth months. Additionally, a difference in the incidence of AD appeared to occur at 6 months of age. We presented Fig 1 to explain why we started this study on the association between seasonal meteorological factors and the incidence of AD. 

Urine and blood from a subset of participants were sampled. Vitamin D, thyroid-stimulating hormone, thyroxine, triiodothyronine, and hemoglobin A1c concentrations were among various items measured (Sekiyama, M. et al. Study design and participants’ profile in the Sub-Cohort Study in the Japan Environment and Children’s Study (JECS). Journal of Epidemiology 2020, JE20200448. Online first). Because we are not permitted to use the data of biospecimens in this study, we were not able to use these data in the analysis. The values of biospecimens are used as exposures or outcomes of epidemiological studies. A few studies using these data have been published (e.g., Ma, C. et al. Association of prenatal exposure to cadmium with neurodevelopment in children at 2 years of age: The Japan Environment and Children's Study. Environ Int 2021, 156, 106762). Household environmental measurements from the JECS have also been published (Iwai-Shimada, M. et al. Questionnaire results on exposure characteristics of pregnant women participating in the Japan Environment and Children Study (JECS). Environmental health and preventive medicine 2018, 23, 45). Although computational simulation was performed at the study core center, we could not find the published information. Therefore, we have deleted the information about computational simulation from our study. The following text was added to the manuscript:

Line 71: “Exposure to environmental factors was assessed by chemical analyses of biospecimens and household environment measurements using monitoring data and questionnaires.”

We extended our analyses to examine the seasonality of the incidence of AD among climate conditions as follows. First, the Kaplan–Meier curve showed that the neonatal period until 6 months of age determined the risk of AD until 3 years of age. Second, in the time-to-event data, we compared the risk per unit of standard deviation among the conditions using the Cox proportional hazard model to determine which climate condition is involved in the incidence of AD because the units of climate conditions are different. Third, because temperature and vapor pressure rather than relative humidity were candidates involved in the incidence of AD, we compared the risk of AD by high/low mean temperature, maximum temperature, minimum temperature, and vapor pressure for 6 months in the model. 

The latitude was plotted based on the birthplace of the studied children. We evaluated Pearson’s correlation coefficient to determine if the latitude is associated with the incidence of AD. If there is an association, latitude-associated climate conditions, including mean temperature and vapor pressure, may be involved in the incidence of AD. Notably, this was an ecological study, and the evidence level was weak. Although there was no significant association, we attempted to identify the reason for the association between the season of birth and the incidence of AD. We have added the reason why we performed this analysis to the Materials and Methods section and explained the results. 

Line 110 in the Materials and Methods section: “Finally, we plotted the latitude and the accumulated incidence of AD at 3 years of age among the birthplaces. Although there were 19 different prefectures, a small subset of data in one prefecture was treated as belonging to data of another prefecture. Therefore, we plotted the latitude for 18 prefectural meteorological observatories. In these plots, we calculated Pearson’s correlation coefficients and p values. In this analysis of our ecological study, if there was a climate condition-associated incidence of AD, we attempted to determine whether it was simply attributable to latitude.” 

Line 171 in the Results section: “We did not find a significant relationship between the latitude and the incidence of AD in this ecological study.”

Line 222 in the Discussion section: “Vapor pressure and temperature are likely to be high if the latitude of the residence is low. Previous studies and our ecological analyses were not able to determine the climate condition responsible for the risk of AD.”

The use of two-sided p values indicated that a two-tailed test was used. We have revised the description of p values as follows: 

Line 116 in the Materials and Methods section: “A p value of <0.05 in two-tailed tests was considered to indicate a significant difference.”

5. Results. Table 1 shows no statistics because there was no difference between the parameters? What does Ref mean as a reference in Table 3. 

We have added asterisks to Table 1 to indicate statistical significance. In Table 3, “Ref” means “reference”. We apologize for not defining this abbreviation and have spelled it out instead. 

6. Discussion. Be specific why any temperature (Low, high?) and vapor pressure are associated with a higher incidence of AD. Is a high heat or cold temperature associated with a likelihood of office visits for AD? 

We thank the reviewer for the comment. We have added explanations about the association between a low temperature and a high incidence of AD. 

Line 179: “When we investigated the association of AD with the climate condition, while avoiding confounding, a low vapor pressure was most strongly associated with a higher incidence of AD in childhood.”

Line 213: “A high temperature enables air to contain more water, and therefore this could increase vapor pressure.”

Line 230: “These definitions mean that vapor pressure rather than humidity can better indicate the moisture in air and may be directly associated with skin moisture. This may be the reason why vapor pressure was more strongly associated with the incidence of AD than humidity in this study.”

7. Conclusion. Make also a statement on the results on temperature and humidity association with higher incidences of AD.

We thank the reviewer for the comment. We mentioned in the Discussion section that among the climate conditions, a low vapor pressure, rather than a low temperature or low humidity, is the most strongly associated with a high incidence of AD. We have modified the summary and conclusion accordingly. 

Line 179: “When we investigated the association of AD with the climate condition, while avoiding confounding, a low vapor pressure was most strongly associated with a higher incidence of AD in childhood. Temperature, maximum temperature, and minimum temperature were also associated with a higher incidence of AD.” 

Line 266: “Among climate conditions, such as temperature and relative humidity, a low vapor pressure is the most strongly associated with a high incidence of AD in Japan. Paying attention to vapor pressure may reduce the incidence of AD in neonates who are born between October and December.”

Responses to comments from reviewer #1

This manuscript follows up on prior results indicating that in Japanese individuals a birth date between October to December correlates with increased risk for atopic dermatitis. The current work tries to tease out the features of seasonality that are responsible. The work is interesting and deserves publication.

We thank the reviewer for the comments. We have revised our manuscript accordingly. 

1. The authors should state limitations of their conclusions in more detail. 

We have added more details for the limitations (see below). We hope that this paragraph now sufficiently explains the limitations. 

Line 254: “Our study had the following limitations. First, the indoor condition was not measured in this study. We could not exclude the effect of a household environment with the use of cleaners, bleaches, and detergents. Second, our results are limited because we did not detect Staphylococcus aureus in the skin [24], which could contribute to an increased risk of AD. Third, dichotomizations of a long/short sunshine duration and high/low humidity were determined by single cut-off values. Observational results could have varied, depending on the cut-off values. Fourth, the incidence of AD was reported by caregivers based on a physician’s diagnosis. Therefore, there could have been recall bias of the caregivers. Fifth, physicians who diagnosed children included specialists of AD and non-specialists. Non-specialist physicians might have underdiagnosed AD in infancy because this diagnosis can cause stigma to children and caregivers [25]. Sixth, short-term factors, such as yellow sand from China and seasonal pollen in the air, were not considered in this study. Studies have suggested that these factors may worsen dermatitis [26, 27].”

2. The sentence about a randomized controlled trial showing prevention of AD by use of moisturizer in infants <1year needs to be modified, reflecting the fact that there are multiple studies on this subject with conflicting results.

We thank the reviewer for the suggestion. We have revised the text in accordance with the reviewer’s advice. 

Line 245: “A meta-analysis showed that emollient use in infants younger than 6 months was significantly effective for preventing AD, while there were conflicting results on its effectiveness in infants aged between 6 and 12 months [21].”

We hope that our responses to the Editor’s and reviewer’s comments and the corresponding manuscript revisions have addressed the main points raised. We are grateful for the helpful suggestions, and we hope that our manuscript is now suitable for publication in PLOS ONE.

Yours sincerely,

Hiroshi Yokomichi 

Department of Health Sciences, University of Yamanashi,

1110 Shimokato, Chuo City, Yamanashi, 4093898, Japan 

E-mail: hyokomichi@yamanashi.ac.jp

Phone: +81 55 273 9569

Fax: +81 55 273 7882

---

## [Editor Report · Decision Letter 1]

25 Apr 2022

Association of the incidence of atopic dermatitis until 3 years old with climate conditions in the first 6 months of life: Japan Environment and Children’s Study (JECS)

PONE-D-22-05326R1

Dear Dr. Yokomichi,

We’re pleased to inform you that your manuscript has been judged scientifically suitable for publication and will be formally accepted for publication once it meets all outstanding technical requirements.

Kind regards,

Svetlana P. Chapoval

Academic Editor

PLOS ONE

---

## [Editor Report · Acceptance letter]

28 Apr 2022

PONE-D-22-05326R1 

Association of the incidence of atopic dermatitis until 3 years old with climate conditions in the first 6 months of life: Japan Environment and Children’s Study (JECS) 

Dear Dr. Yokomichi:

I'm pleased to inform you that your manuscript has been deemed suitable for publication in PLOS ONE. Congratulations! Your manuscript is now with our production department. 

Kind regards, 

on behalf of

Dr. Svetlana P. Chapoval 

Academic Editor

PLOS ONE